# Deletion of the *Spata3* Gene Induces Sperm Alterations and In Vitro Hypofertility in Mice

**DOI:** 10.3390/ijms22041959

**Published:** 2021-02-16

**Authors:** Marie-Sophie Girault, Sophie Dupuis, Côme Ialy-Radio, Laurence Stouvenel, Cécile Viollet, Rémi Pierre, Maryline Favier, Ahmed Ziyyat, Sandrine Barbaux

**Affiliations:** 1Institut Cochin, Université de Paris, INSERM, CNRS, F-75014 Paris, France; marie-sophie.girault@inserm.fr (M.-S.G.); sophie.dupuis@inserm.fr (S.D.); come.ialy-radio@inserm.fr (C.I.-R.); laurence.stouvenel@inserm.fr (L.S.); violletcecile@gmail.com (C.V.); remi.pierre@inserm.fr (R.P.); maryline.favier@inserm.fr (M.F.); aziyyat@yahoo.fr (A.Z.); 2Service d’Histologie, d’Embryologie, Biologie de la Reproduction, AP-HP, Hôpital Cochin, F-75014 Paris, France

**Keywords:** *Spata3*, infertility, acrosome, spermatogenesis, sperm, mouse model, lipid droplets

## Abstract

Thanks to the analysis of an Interspecific Recombinant Congenic Strain (IRCS), we previously defined the *Mafq1* quantitative trait locus as an interval on mouse Chromosome 1 associated with male hypofertility and ultrastructural abnormalities. We identified the Spermatogenesis associated protein 3 gene (*Spata3* or *Tsarg1*) as a pertinent candidate within the *Mafq1* locus and performed the CRISPR-Cas9 mediated complete deletion of the gene to investigate its function. Male mice deleted for *Spata3* were normally fertile in vivo but exhibited a drastic reduction of efficiency in in vitro fertilization assays. Mobility parameters were normal but ultrastructural analyses revealed acrosome defects and an overabundance of lipids droplets in cytoplasmic remnants. The deletion of the *Spata3* gene reproduces therefore partially the phenotype of the hypofertile IRCS strain.

## 1. Introduction

Genes necessary for testis function, particularly in post-meiotic germ cells, are extremely numerous [1]. However, the function and character of dispensability of these genes is only partially known. Their responsibility in clinical cases of human male infertility is therefore also only partially deciphered. 

For example, globozoospermia, a very rare form of male infertility characterized by round-shaped sperm heads that present a complete lack or an abnormal setup of their acrosome, can be explained by alterations of the *DPY19L2* gene in about 60% of reported cases [2,3]. Some rare case reports have also revealed mutations of the *SPATA16*, *ZPBP1* and *PICK1* genes [4,5,6]. But the defect responsible for some remaining clinical cases is not yet identified and the biogenesis of the acrosome is still not completely elucidated. Animal models can help with suggesting some other genes involved in the process. 

Recently, we reported the analysis of a mouse model exhibiting a phenotype of spermatogenesis alterations including anomalies of the acrosome. We identified in an Interspecific Recombinant Congenic Strain (IRCS) a region of mouse chromosome 1 whose presence in a *Mus spretus* version within a *Mus musculus domesticus* (C57Bl/6) background is associated with signs reminiscent of globozoospermia. This quantitative trait locus (QTL), named *Mafq1* for male fertility QTL chromosome 1, is associated with an interval of approximately 4 Mb between markers *D1Mit438* and *D1Mit305* [7].

This specific interval contains more than 70 coding genes. Among them, we concentrated our research on the *Spata3* gene (Spermatogenesis associated protein 3, also known as Testis and Spermatogenesis cell Apoptosis Related Gene 1 or *Tsarg1*) [8,9]. We consider it as one of the best candidates to explain the hypofertility phenotype of the Rc3 strain, on the basis of various arguments.

*Spata3* is expressed specifically in the testis both in mice and humans. Precisely, its expression can be detected during spermatogenesis from the round spermatid stage to mature sperm [7]. *Spata3* is conserved in many mammalian species, with mouse and human proteins sharing 55% of their identity. 

No data are available on the function of the SPATA3 protein, the SPATA family being a catalogue of proteins sharing an expression in the testis without any similarity of sequence neither function. Furthermore, no specific functional domain can be predicted from its sequence and no mouse model is available to determine its role. However, the human SPATA3 gene has been reported to be among the most down-regulated genes in the testis of infertile men diagnosed with nonobstructive azoospermia [10]. Rare variants of this gene were also associated with prostate cancer [11].

Comparison of the sequence of the *Spata3* gene between the *Mus spretus* and the C57Bl/6 versions highlights many variations: 14 missense changes in the different isoforms and at least 16 UTR variants (Appendix A). These variations could be responsible for important modifications of the structure and/or the function of the *spretus* SPATA3 protein compared to the C57Bl/6 version. This *spretus* version of the protein could therefore be inapt to interact/bind/function with any receptor/cofactor/partner in the C57Bl/6 version in the IRCS model to accomplish its normal activity, resulting in the hypofertility phenotype observed in the Rc3 IRCS male mice [7]. 

We therefore decided to produce a mouse line invalidated for the *Spata3* gene in order to decipher the function of this gene. Though male mice lacking *Spata3* expression are fully fertile in vivo, their fertility in vitro is drastically affected and some ultrastructural anomalies can be observed in sperm samples.

## 2. Results

### 2.1. Spata3 Is Expressed in Sperm and Localized on the Acrosome

We used a specific antibody to detect SPATA3 in testis slides. SPATA3 is expressed from the round spermatid stage to mature sperm. It is specifically localized at the acrosome region (Figure 1a–d). No signal was obtained when secondary antibody was used alone (Figure 1e,f). We also confirmed its presence at the mRNA level by RT-PCR, as shown in Figure 2b. Quantitative single-cell RNA sequencing analyses confirm a germ-cell specific expression from the round spermatid stage in humans and mice (Reprogenomics Viewer) [12,13].

### 2.2. Spata3 Gene Deleted Male Mice Are Fertile In Vivo

Mice invalidated for the *Spata3* gene were generated by microinjecting embryos with RNA guides targeting upstream of the first coding exon (exon 1bis) and downstream of the last coding exon (exon 3) (Figure 2a). Two different though very similar lines encompassing an 8547 nt deletion and an 8507 nt deletion of the gene locus, respectively, were selected and maintained. Identification of animals presenting a deletion of *Spata3* was obtained by PCR amplification and sequencing of the locus as illustrated in Figure 2c. Different crossings were performed to produce homozygous knock-out animals (KO). Females *Spata3* KO were normally fertile. As the antibody previously used for immunofluorescence detection (Figure 1) is no longer commercialized and no other validated antibody is available for Western blotting, the absence of expression of the *Spata3* gene could only be checked by RT-PCR. No amplification of *Spata3* could be observed in the testis of KO males compared to wild-type (WT) samples whereas the cDNA of a testis-specific gene, *Prm1*, could be detected in both samples (Figure 2b) [14]. *Spata3* KO males exhibited a normal sexual behavior and after mating with WT females, could sire offspring (Figure 3a). The average litter size was not different when pups were born from KO males (7.1 ± 0.9; *n* = 9) or from WT males (6.7 ± 0.5; *n* = 10, *p* = 0.71). This result suggests that in vivo the *Spata3* gene is dispensable for sperm function and fertility. Similar results were obtained with the two KO strains but only one representative strain is shown. 

### 2.3. Spata3 Gene Deleted Male Mice Have a Reduced Fertility In Vitro

In vitro fertilization assays were set up to further analyze the sperm quality. The fertilization rate (FR: number of fertilized oocytes among the total) of cumulus-intact WT oocytes was 34.9 ± 3.9 % (*n* = 147) when WT sperm was used and 7.3 ± 2.1% (*n* = 150) when the sperm was obtained from KO males (*p* < 0.0001) (Figure 3b). When zona-free oocytes from WT females were inseminated with sperm from WT or KO males, the fertilization index (FI: the average number of fused sperms per oocyte) was 0.92 ± 0.06 (*n* = 108) and 0.35 ± 0.04 (*n* = 118), respectively (*p* < 0.0001) (Figure 3c). These observations confirm that *Spata3* is not essential for in vivo fertilization but that in vitro, its absence affects the successful process of fertilization.

### 2.4. Sperm from Spata3 Gene Deleted Male Mice Have Structural Anomalies

In order to identify why sperm from *Spata3* KO mice were not as efficient to fertilize in vitro, we decided to characterize it. Histology of the testes of KO males showed no abnormalities (Figure 4a). Observation of sperm under a light microscope failed to show any obvious defect (Figure 4b). Counts from epididymal sperm were not different between WT and KO males (Figure 4c). Similarly, spontaneous acrosome reaction occurred at the same level in WT and KO males after sperm capacitation (Figure 4d). Analyses using the Computer Aided Sperm Analysis (CASA) system did not reveal any difference between the control and KO sperm for any parameters, including curvilinear velocity (VCL), beat/cross frequency (BCF) and amplitude of lateral head displacement (ALH), as shown in Figure 4e,f. 

Sperm samples were analyzed by electronic microscopy in order to reveal subtle morphological and ultrastructural defects that could not be visible through light microscopy and that could explain the in vitro hypofertility phenotype. Some abnormal sperm could be observed, while the acrosome seemed to be partially or completely detached from the nucleus. In some extreme cases, sperm could also look “inflated”, showing an asymmetrical acrosome, completely untied from the nucleus, as exemplified in Figure 5a. A count of this particular phenotype revealed a greater abundance in KO (15%) compared to WT (3%) samples (Figure 5c). Some other sperm showed only one or the other defect, that were not considered in the previous quantitative evaluation.

In addition, some dark droplets were visible, more frequently in KO compared to WT samples (Figure 5b). These droplets, suspected to contain lipids, are present within cytoplasmic remnants located around the midpiece of the flagellum, as attested to by the nearby presence of mitochondria.

In order to better characterize a putative metabolic defect resulting in such droplets, we used an Oil Red O staining on WT and KO sperm (Figure 6a). Up to 13.4% and 9.4% of KO sperm were positive for lipid droplets, before and after capacitation, respectively, whereas less than 0.5% of WT sperm presented staining in both conditions (Figure 6b).

## 3. Discussion

In order to understand which gene within the *Mafq1* QTL interval was responsible for the hypofertility phenotype observed in the Rc3 IRCS mice [7], we explored the list of candidate genes present in the interval. *Spata3* matched all potential criterion of selection: the specific expression profile, regarding both the timing and location in differentiating sperm cells, the evolutionary conservation, the high number and predicted impact of missense changes between B6 and *spretus* SPATA3 proteins. The lack of information about its function left all hypotheses open. We therefore decided to generate a *Spata3* null mouse to determine if the invalidation of the *Spata3* gene could mimic the phenotype of the Rc3 IRCS mouse.

When this project was initiated, the reported structure of the *Spata3* gene was the one reproduced in Figure 2a, as isoform 1, including 3 coding exons (Exons 1, 2 and 3) and additional variable 3′UTR exons. A CRISPR/Cas9 strategy had been developed to target and affect exon 1 with RNA guides, therefore compromising the full ORF. Two independent KO lines were produced but they were found to be fully fertile in vivo (data not shown). 

However, in 2018, databases revealed the existence of so far unsuspected isoforms of the gene. First, an additional isoform starts upstream of the reported translation start, at an ATG located 303 bp earlier, in phase (“exon 1 upstream”, Figure 2a, isoform 2). Second, a new alternative isoform includes a new upstream first exon (“exon 1 bis”, Figure 2a, isoform 3), followed by the classic coding exons 2 and 3, in phase. This new messenger therefore encodes a protein that shares the last 92 amino acids but completely differs from the initially known form at the *N*-terminus and is not affected by the absence of the “classic” exon 1. This gene organization is conserved, while isoforms 1 and 3 are present in mice and humans. In 2019, Zhou et al. reported new isoforms of the human *SPATA3* gene [15]. They first identified a new isoform skipping exon 2, leading to a protein of 138 amino acids whose C-terminal end is different because of a frameshift, but whose abundance is unknown. They also described an alternative splicing within exon 1, where 27 bp would be spliced away. However, this is rather a documented polymorphism, referenced as rs 72362780, with different alleles of either 2 or 3 repeats of a 27 bp sequence encoding a QQPSPESTP motif, being almost equally frequent in the human population. 

The initial knock-out was therefore considered as a partial KO, suppressing only some of the possible isoforms and probably leaving some not negligible residual SPATA3 activity. Another knock-out strategy was therefore adopted to delete all known coding exons. RNA guides located upstream exon 1bis and downstream exon 3 were used to generate a complete gene deletion of about 8 kb. Two independent KO lines were obtained for this complete invalidation that gave identical results. Sequencing of the locus confirmed the deletion of the coding exons and validated the precise breakpoints. The lack of expression of the *Spata3* cDNA was confirmed by RT-PCR, in the absence of an efficient antibody to detect the SPATA3 protein by Western blot. When mated with WT females, homozygous KO males could sire normally and produce normal litters. The absence of *Spata3* did not reproduce the in vivo hypofertility observed in Rc3 males [7]. However, in IVF assays, sperm lacking *Spata3* expression were less efficient to fertilize WT oocytes. Both the fertilization rate of cumulus-intact oocytes and the fertilization index of zona-free oocytes were drastically reduced. This difference of phenotype between in vivo and in vitro conditions has already been observed. It is probably due to the differences between an optimal in vivo fertilization and the artificial context of IVF where some parameters are not ideally reproduced. Thus, IVF could exacerbate a defect that is compensated for under physiological conditions. This observation has already been encountered for the sperm specific *Itgb1* KO [16] and for the mouse heterozygous for a *Spaca6* deletion [17]. This reduction of IVF efficiency was also detected in Rc3 males.

No gross abnormality could be detected in the testicular histology neither in sperm counts, suggesting a normal spermatogenesis. The sperm movement parameters explored by CASA were not affected. A reduction of the motile fraction had been observed in Rc3 males but here the decrease in IVF yield cannot be explained by a reduced motility of sperm. Sperm ultrastructure was explored by electronic microscopy to observe more subtle sperm alterations, as were reported in the Rc3 IRCS model [7]. Some abnormal sperm could be observed, mainly affecting the acrosome. As observed in the Rc3 line, an acrosome detached from the nucleus is more frequently reported in KO than in WT sperm. The acrosome sometimes looks out of shape, far from the expected symmetric cap aspect. It is very likely that the totally abnormal appearance of the sperm shown in Figure 5a is not really the ejaculated sperm phenotype but rather the result of manipulations which further damage the structure of these KO sperm compared to WT sperm. It is anyway a sign of a particular fragility of KO sperm. In addition, dark droplets in persistent cytoplasmic remnants were also visible. An Oil Red O coloration was performed and confirmed the lipid nature of these droplets contents and that they are much more present in KO sperm. These cytoplasmic bodies should have been removed by Sertoli cells in the last steps of spermiogenesis. It is not clear if the phagocytosis process is altered in KO testes, but SPATA3 has been shown to be expressed in germ cells, not in Sertoli cells. Alternatively, a dysfunction of lipid metabolism could be present in KO sperm, leading to an excess of lipid droplets that fail to be correctly phagocyted by Sertoli cells. Why this process of lipids accumulation happens will need more analyses. It is difficult to estimate exactly what percentage of sperm is abnormal, as, more than 10% showed lipid droplets, 15% appeared fully disorganized at the acrosome level, and some others showed milder defects.

In conclusion, male mice lacking the expression of the *Spata3* gene are fertile in vivo, thanks to the optimal context of sperm selection and gamete interaction. They are nevertheless hypofertile in vitro. Moreover, they have a higher frequency of abnormal sperm. Specifically, they have more abnormal acrosomes that tend to untie from the nucleus and more cytoplasmic bodies showing an imperfect spermiogenesis and spermiation. 

The *Spata3* KO, though modestly affecting sperm capacities, is therefore an incomplete reproduction of the Rc3 IRCS mouse line, that was considered as a model of partial globozoospermia. Two hypotheses can be proposed. It is possible that the complete suppression of SPATA3 protein expression in the invalidated mouse on one side and the expression of a modified SPATA3 protein in the IRCS on the other have different impacts on sperm function. The presence of a deleterious protein could be more damaging than its absence, if for example, some compensatory mechanisms can set up. Some of the polymorphisms existing between *spretus* and C57Bl/6 strains are indeed individually predicted to affect protein function. However, no prediction model can aggregate the effect of the 14 non-synonymous changes on the integrity of the different protein isoforms, especially on a protein of unknown function. The alternative hypothesis is that another gene within the *Mafq1* interval contributes to the complex phenotype of the Rc3 males. One or more gene(s) present in the QTL interval and exhibiting variations in the *spretus* strain compared to the C57Bl/6 strain, likely missense changes affecting protein function but also possibly regulatory variants affecting protein levels, could also be dysfunctional in the Rc3 line. The cumulative effects of two or more disturbed genes could be necessary to affect more drastically sperm function, either directly if proteins are acting together or indirectly if they have different functions whose anomalies add at different stages or pathways to generate the defects profile at the end. A careful re-analysis of the identity and characteristics of the genes present in the Rc3 interval could suggest other candidates that could participate in the phenotype. 

On the human side, SPATA3 and other potential candidate genes from the QTL region should be screened in infertile/hypofertile patients with signs reminiscent of globozoospermia. Furthermore, genes identified in this QTL region will be particularly closely analyzed in our future whole exome sequencing studies of infertile patients.

## 4. Materials and Methods 

### 4.1. Animals

All animal experiments were performed in agreement with national guidelines for the care and use of laboratory animals. Experimentation was approved by the local ethics committee (C2EA-34, Comité d’éthique en matière d’expérimentation animale Paris Descartes) and the governmental ethical review committees via the APAFiS Application (Autorisation de projet utilisant des animaux à des fins scientifiques), under the registration reference APAFIS #14124-2017072510448522 v26, A. Ziyyat (10/102018-10/10/2023).

Mice were purchased from Janvier Labs (Le Genest-Saint-Isle, France). All animals were maintained at the animal facility of the Cochin Institute (Paris) at a stable temperature (21–23 °C) and 14 h light/10 h dark photoperiods, with free access to food and water.

### 4.2. Generation of Transgenic Mice

The Crispr/Cas9 approaches were performed at the Transgenesis and Homologous Recombination platform (PRHTEC) of the Cochin Institute, Paris, France. RNA guides were selected, produced by TACGENE (Muséum d’Histoire Naturelle, Paris, France) and tested in vitro (targeting upstream exon 1 bis: ACCGGACAGACATGGAACAG; GCTGGGACCCCTGTTAGCCAA; GTTGTGACATGCTCCAGGGAA- targeting downstream exon 3: GGACCCCAGGCCCACCGTC; GTAGAGAAGACTATCTCAAGG; GGCCCACCGTCCGGCAGGT). 

They were preincubated with Cas9 and then microinjected into fertilized C57Bl/6J oocytes. 

Embryos that had passed to the 2-cell stage were transferred into the oviducts of pseudopregnant females.

### 4.3. PCR and RT-PCR

Genotyping was performed on DNA extracted from tail biopsies (NucleoSpin^®^ Tissue, Macherey-Nagel, Düren, Germany) using the GoTaq Flexi polymerase (Promega, Madison, WI, USA) under standard PCR conditions. Primers are listed in Appendix A. The integrity of the areas targeted by the guides was tested to detect WT alleles. A large amplification encompassing the whole locus was also performed to detect deleted alleles. 

RNA was obtained by TRIzol (Thermo Fischer Scientific, Waltham, MA, USA) extraction from WT and KO testes. Five micrograms of total RNA were treated with DNAse I (Promega) and retrotranscribed using the MMLV reverse transcriptase (Thermo Fischer Scientific) following manufacturer’s instructions. RT-PCR was performed using the GoTaq Flexi polymerase to test the expression of *Spata3* and a reference gene as normalization control.

### 4.4. In Vivo and In Vitro Fertility Assessment

Mice, aged from 8 to 12 weeks, were housed as one male and one female per cage. WT females were mated overnight with WT males or homozygous KO males and checked for vaginal plugs the next day. For each group, litter sizes were assessed. 

Wild-type 5–8 weeks-old female mice were superovulated with a 10 IU pregnant mare serum gonadotrophin (PMSG) injection, then 48 h later with a 10 IU human chorionic gonadotrophin (hCG) injection (Intervet, Beaucouzé, France). The next day, 14 h post-injection, animals were sacrificed by cervical dislocation. Cumuli oophori were retrieved by tearing the ampulla wall of the oviduct, placed in Ferticult medium (FertiPro, Beernem, Belgium) supplemented with 3% bovine serum albumin (BSA), and maintained at 37 °C under 5% CO_2_ in air, under mineral oil (Sigma Aldrich, Saint Quentin Fallavier, France).

For zona-free IVF assay, oocytes were released from the cumulus cells by 3–5 min incubation at 37 °C with hyaluronidase (Sigma Aldrich) in M2 medium (Sigma Aldrich), rinsed and kept in Ferticult medium, BSA 3% at 37 °C under a 5% CO_2_ atmosphere with mineral oil (Sigma Aldrich). The zona pellucida was then dissolved with acidic Tyrode’s (AT) solution (pH 2.5, Sigma Aldrich) under visual monitoring. Zona-free eggs were rapidly rinsed in a Ferticult medium, BSA 3% and kept at 37 °C under a 5% CO_2_ atmosphere for 2 to 3 h to recover.

Mouse spermatozoa were obtained from the cauda epididymis of control or KO male mice aged 8 to 13-weeks. They were capacitated at 37 °C under a 5% CO_2_ for 90 min in a 500 µL drop of Ferticult medium supplemented with 3% BSA (30 mg/mL), under mineral oil.

Cumulus-intact or zona-free oocytes were inseminated in a 100 µL drop of Ferticult medium, BSA 3% with capacitated spermatozoa at a final concentration of 1×10^6^/mL or 1×10^5^/mL, respectively, for 3 h. They were then washed and directly mounted in Vectashield/DAPI (Vector Laboratories, Burlingame, CA, USA) for observation under UV light (Zeiss Axioskop 20 microscope, Marly le Roi, France). Oocytes were considered fertilized when a fluorescent decondensed sperm head could be visualized within their cytoplasm.

### 4.5. Histology, Staining and Immunostaining of Sperm and Testes

Testes from adult WT and KO males were collected. They were fixed overnight in 4% (*w*/*v*) paraformaldehyde in PBS, dehydrated and embedded in paraffin. Serial 5 µm thick sections were cut on a microtom and stored at room temperature. Testis sections were rehydrated before Hematoxylin-eosin (H&E) staining.

For Immunofluorescence staining, slides were permeabilized during 10 min in PBS 1× and with triton 0.2% at RT, and were then saturated for 30 min in PBS, BSA 3% and Goat serum 10%. Slides were incubated for 1 h in a polyclonal anti-TSARG1 antibody at 1:20 in PBS (T-18, sc-67901, Santa Cruz Biotechnology, Dallas, TX; reference no longer available), washed three times in PBS, BSA 1%. DAPI (5 µg/mL, Sigma Aldrich) and secondary antibody (anti-Goat-Alexa 488 at 10 µg/mL, ThermoFischer Scientific) were used at RT during 1 h, followed by three washes and slides were mounted with the Dako Fluorescent mounting medium (Agilent, Santa Clara, CA, USA). 

For Papanicolaou staining, sperm were retrieved from cauda epididymes in Ferticult medium and spread onto a Superfrost Plus slide (ThermoFischer Scientific). Sperm cells were fixed by incubation with PBS, paraformaldehyde 4% for 10 min and stained following the Papanicolaou protocol (Hematoxylin, OG6, EA50). Briefly, the slides were washed in 95% ethanol and inserted in Harris hematoxylin for 3 min for nucleus counterstaining, and washed, stained with OG-6 dye (RAL diagnostics 05.12013, Martillac, France) and with EA-50 (RAL diagnostics 05.12019, Martillac, France), Then the slides were dehydrated (95% ethanol absolute ethanol and xylene) and mounted with a permanent mounting medium.

Freshly recovered or capacitated sperm were washed in PBS containing 1% BSA, centrifuged at 300 g for 5 min and immediately fixed in 4% Paraformaldehyde (Electron Microscopy Sciences, PA, USA) in PBS-1% BSA for 10 min at room temperature. In order to detect the sperm acrosomal status, after washing, fixed spermatozoa were stained with FITC-conjugated lectin Pisum Sativum Agglutinin (PSA; 25 μg/mL in PBS, Sigma Aldrich) for 15 min. After repeated washing with PBS-BSA 1%, a drop of sperm suspension was smeared on a slide, air-dried, mounted with Vectashield-DAPI. Detection was performed using a Nikon Eclipse E600 microscope Zeiss Axiophot epifluorescence microscope and images were digitally acquired with a camera (Coolpix 4500, Nikon, Champigny sur Marne, France).

For Oil Red O (ORO) staining, a drop of freshly recovered or capacitated sperm was smeared on a slide, air-dried and immediately fixed in PFA 4% and stored at −80 °C until the day of use. Slides were post-fixed for 10 min in PFA 4%, then rinsed and quenched with NH_4_Cl 50 mM for 10 min. After rinsing, cells were permeabilized for 4 min with triton X-100 0.5%. Finally, cells were stained by an incubation of one hour with Oil-Red-O (Sigma Aldrich) and counter-stained with hematoxylin of Mayer. Lipids appeared as red droplets and nuclei appeared to be blue. The amount of lipid droplets by ORO staining in sperm cells was quantified using a light microscope.

### 4.6. Mouse Sperm Motility Analysis

Sperm motility was assessed by Computer Aided Sperm Analysis (CASA) using CEROS II apparatus (Hamilton Thorne, Beverly, MA, USA). Briefly, mouse spermatozoa were obtained from the cauda epididymis of *Spata3* KO or WT C57BL6/J mice (8 to 10 weeks old) and capacitated at 37 °C under 5% CO_2_ for 90 min in a 500 μL drop of Ferticult medium supplemented with 3% BSA, under mineral oil. The movements of at least 500 sperm cells per sample were analyzed in 20 µm chambers (Leja Products B.V., Nieuw-Vennep, Netherlands) with Zeiss AX10 Lab. A1 microscope (10× objective), using HT CASAII software.

The settings were as follows: acquisition rate, 60 Hz; number of frames, 45; minimum head brightness, 175; minimum tail brightness, 80; minimum head size, 10 µm^2^; minimum elongation gate, 1%; maximum elongation gate, 100%; objective magnification factor, 1.2.

The principal motility parameters measured were: curvilinear velocity (VCL), average path velocity (VAP), straight-line velocity (VSL), beat/cross frequency (BCF), amplitude of lateral head displacement (ALH).

### 4.7. Electronic Microscopy

Mouse spermatozoa from males were prepared as described above (IVF) and fixed by incubation in PBS supplemented with 2% glutaraldehyde (Grade I, Sigma) for 2 h at room temperature. Samples were washed twice in PBS and post-fixed by incubation with 1% osmium tetroxide (Electron Microscopy Sciences), after which they were dehydrated by immersion in a graded series of alcohol solutions and embedded in Epon resin (Polysciences Inc.,Warrington, PA, USA). Ultra-thin sections (90 nm) were cut with a Reichert Ultracut S ultramicrotome (Reichert-Jung AG, Wien, Austria) and were then stained with uranyl acetate and lead citrate. Sections were analyzed with a JEOL 1011 microscope and digital images were acquired with a Gatan Erlangshen CCD camera and Digital Micrograph software. The integrity of sperm organelles was checked including the acrosome, nucleus, flagellum, etc. Acrosomes with a space between the nuclear membrane and the inner acrosomal membrane were considered abnormal. Acrosomes with membranes in close contact were considered normal. The presence of residual bodies and lipid droplets was recorded.

### 4.8. Statistical Analysis

Results are expressed as mean ± SEM of at least three independent experiments. For statistical analysis, *t*-Test was performed using GraphPad Prism version 7.00 for Windows, (GraphPad Software, La Jolla, CA, USA). Differences were considered statistically significant when *p* < 0.05. When data were not normally distributed, a Welch’s correction was applied.

## Figures and Tables

**Figure 1 ijms-22-01959-f001:**
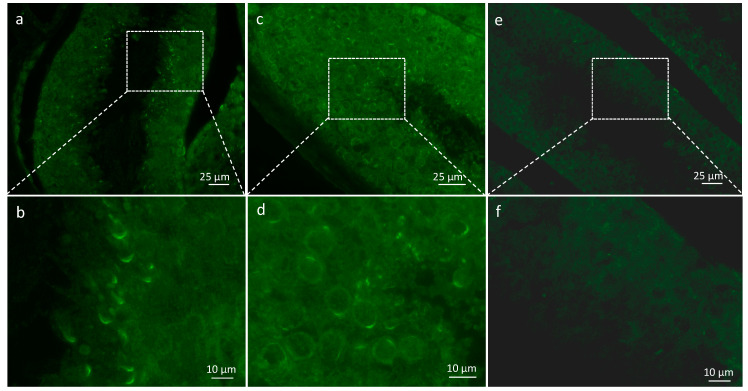
Presence of SPATA3 at the acrosome level. (**a**,**c**) Immunostaining for SPATA3 detection. (**e**) shows negative control using secondary antibody alone and (**b**,**d**,**f**) show a higher magnification of regions framed in (**a**,**c**,**e**), respectively.

**Figure 2 ijms-22-01959-f002:**
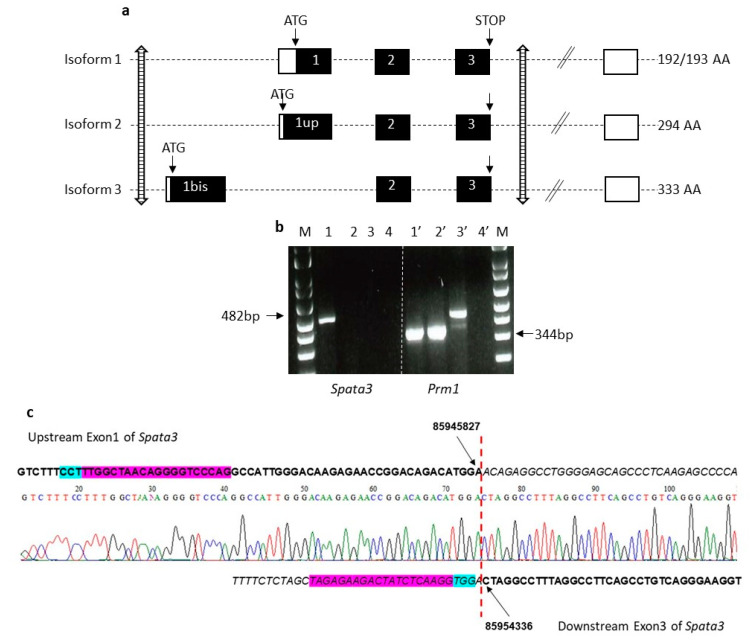
Strategy of deletion of the mouse *Spata3* gene. (**a**). Map of the mouse *Spata3* gene locus. The organization of the mouse *Spata3* gene is represented with the main exons (coding in black boxes, non-coding in white boxes). Isoforms 1, 2 and 3 represent the major isoforms. The position of the sites targeted by RNA guides is marked by striped arrows. Genome version GRCm38.p6. (**b**). Expression of *Spata3* and *Prm1* genes in testis samples. 1 and 1′, wild-type (WT) cDNA; 2 and 2′, knock-out animals (KO) cDNA; 3 and 3′, WT genomic DNA; 4 and 4′, negative control; M, molecular weight marker. (**c**). Chromatogram of the deletion site including alignment with the WT sequence. The RNA guides are in purple and the PAM sites in blue. The positions of deletion sites are indicated.

**Figure 3 ijms-22-01959-f003:**
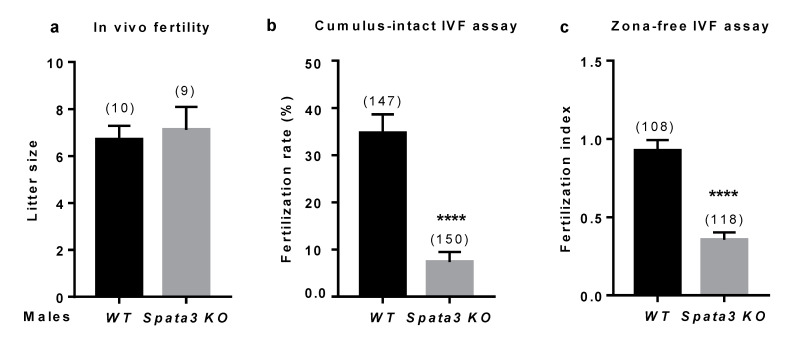
Evaluation of *Spata3* KO male fertility. (**a**). Evaluation of in vivo fertility. WT females were mated with WT or KO males to evaluate their capacity to fertilize in vivo. After an overnight mating, vaginal plugs were checked. Litter size was counted after 3 weeks of gestation. Mean litter size ±SEM. Numbers between brackets indicate the numbers of litters. (**b**). Evaluation of in vitro fertility. The fertilization rate (or mean percentage ±SEM of fertilized eggs) was assessed by cumulus-intact IVF assay at 10^6^ (WT and KO) sperm per ml. It was decreased between oocytes inseminated by KO sperm compared to WT. Numbers between brackets indicate the numbers of oocytes. (**c**). Evaluation of in vitro fertility. The fertilization index (or mean ± SEM of sperm number fused by egg) was assessed by zona-free IVF assay at 10^5^ (WT or KO) sperm per ml. It was decreased between oocytes inseminated by KO sperm compared to WT. Numbers between brackets indicate the numbers of oocytes. Experiments were repeated at least three times. **** *p* < 0.0001.

**Figure 4 ijms-22-01959-f004:**
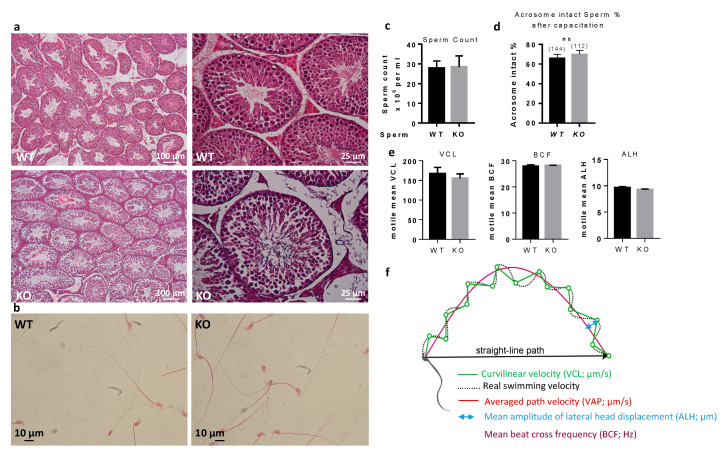
Histology of sperm and testes. (**a**). Adult testes were fixed in paraformaldehyde, sliced and colored with hematoxylin-eosin. (**b**). Papanicolaou staining of WT and KO sperm. (**c**). Sperm counts of WT and KO sperm. (**d**). Spontaneous acrosome reaction counting. There is no difference between WT and KO sperm in terms of acrosome reaction after capacitation. ns, not significant. (**e**). Analyses of sperm parameters by CASA. CASA analyzes of sperm movement parameters from *Spata3* deleted mice showed no difference in comparison to those of WT sperm. As an example, here are presented: curvilinear velocity (VCL), beat/cross frequency (BCF) and amplitude of lateral head displacement (ALH). Data are represented as the mean ± SEM of at least three sperm samples from males of each group. (**f**). Schematic representation of CASA measurements.

**Figure 5 ijms-22-01959-f005:**
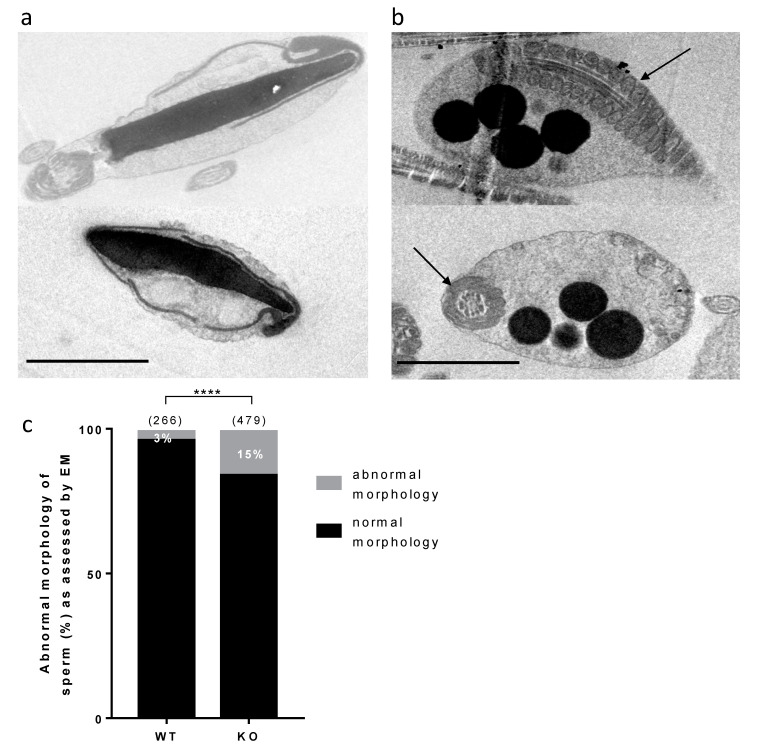
Electronic microscopy analysis of sperm. (**a**). Examples of abnormal sperm heads combining an “inflated” aspect, a detached and asymmetrical acrosome. (**b**). Examples of lipid droplets present in cytoplasmic bodies. The arrows indicate the presence of mitochondria within the flagellum. The scale bars represent 2 μm. (**c**). Counting of sperm showing the specific phenotype exemplified in panel a, among all observed spermatic heads, between WT and KO samples. Numbers between brackets indicate the numbers of sperm heads. **** *p* < 0.0001.

**Figure 6 ijms-22-01959-f006:**
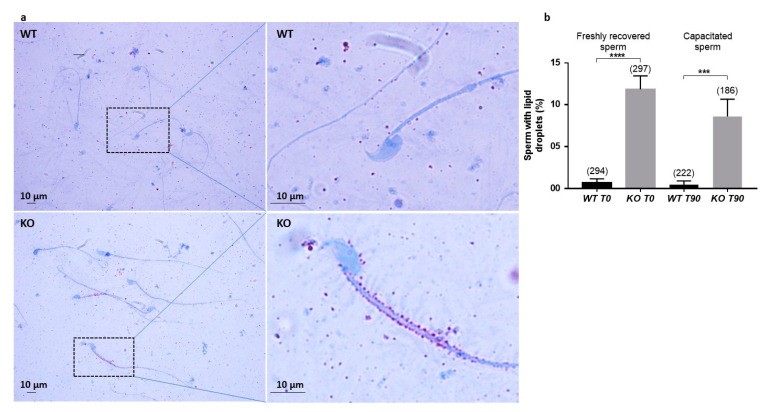
Lipid detection by Oil Red O staining. (**a**). Light microscopy detection of lipid droplets after Oil Red O staining. Right panels show higher magnification of regions framed in left panels. (**b**). Counting of sperm positive for lipid droplets after staining, before (T0) or after (T90) capacitation. Numbers between brackets indicate the counted sperm; *** *p* < 0.001 **** *p* < 0.0001.

## Data Availability

Not applicable.

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
