# Peer review of "Deletion of the *Spata3* Gene Induces Sperm Alterations and In Vitro Hypofertility in Mice"

_ijms, 2021, doi:10.3390/ijms22041959_

Round 1

Reviewer 1 Report

In their current manuscript Girault, Dupius and al. addressed the reproductive phenotype and reproductive performance of the newly prepared Spata3 KO mice model. The major finding of the manuscript is the observed significant decrease in in vitro fertilization assays (despite no significant difference in in vivo fertility) and higher retention of lipids in the sperm cells isolated from KO model. I think this is an interesting manuscript uncovering potential functions of Spata3 gene using KO model but several major points have to be addressed by authors before the decision about the publication can be made.

Major points:

  •   Figure 1: This figure is not convincing. The panel a seems to be DAPI channel combined with overexposed green channel. b-e the signal from acrosomes seems to be classical autofluorescence in the acrosomal area of sperm and not the specific fluorescent signal from secondary antibody. Authors should provide the micrographs from experimental sample/negative control both from the corresponding area with acrosomal autofluorescence to estimate the signal difference. Furthermore, the IHC staining and labelling of the epididymal sperm should be provided. Finally, since authors dispone with KO model, all pictures should be presented in parallel – WT/KO – like in the Figure 4a.

  • Figure 2: Please provide the sequencing results of the KO splicing side with the WTsequence alignment and corresponding KO chromatogram.

  • Figure 1 (updated version) or 2. Please provide the immunoblot figure showing the depletion of Spata3 protein in the KO testicular tissue or sperm protein lysate (WT/KO).

Minor points:

  • Figure 5. Some electron micrographs of the testicular tissue slides demonstrating the formation of the lipid droplets in the tissue context would increase the merit of the figure and whole manuscript.
  • Line 300 please use the term “reference gene” instead of “housekeeping gene”. Prm1 should also not be used as for normalization (as indicated in Supplementary table 2 since its abundance can variates significantly.
  • Authors should more deeply address the issue why such a significant changes are observed in in vitro assays vs in vivo fertility with regard of only around 10% changes in the observed phenotypic parameters (Fig 5c; 6b).

Author Response

We thank reviewer 1 for his/her pertinent comments.

-We completely agree with him/her about the relevance of some additional experiments involving an anti-SPATA3 antibody. Unfortunately, they are not possible.

The experiment shown in figure 1 was done a few years ago when Spata3 was first explored as a candidate gene. Antibodies from Santa Cruz Biotechnology (ref sc-67900 and sc-67901) were used at that time that gave satisfactory results. However, these products are no more available from SCB, neither from any other retailer. We extensively searched for other specific antibodies and tested the few ones on the market:

  • A Proteintech polyclonal antibody that recognizes only the human protein, but not very efficiently in our hands.
  • A Bioss polyclonal antibody supposed to recognize human, mouse and rat SPATA3 protein but showing no band on immunoblot.
  • A Signalway Antibody polyclonal antibody. A careful reading of its description revealed that there is a confusion between SPATA3 (TSARG1) and DNAJB13 (TSARG3). The western blot validation included in the datasheet, showing a band in lung tissue, confirmed the misleading attribution as SPATA3 is testis specific.

There is another Sigma antibody but it is human-specific.

Therefore, we don’t have appropriate tools to perform the immunofluorescence or immunochemistry labeling on KO vs WT sperm and the immunoblot to confirm the absence of protein in the KO. To this goal, we used the RT-PCR experiments.

-We completed figure 1 to include negative controls. On testis sections, autofluorescence is very weak, compared to what happens with sperm samples. Labeling with only the secondary antibody shows no autofluorescence and no specific signal.

-We added in figure 2c the sequencing of the junction resulting from the deletion of the gene.

- We don’t have electron microscopy images from testicular tissue and the process to generate them is very long, particularly in this specific pandemic context. We have other images from lipid droplets in sperm samples, if requested.  

-We changed the word housekeeping gene to reference gene.  It is possible that the level of expression of Prm1 is affected in the KO. Nevertheless, the experiment depicted in figure 2 is not a quantitative evaluation of expression level. Its objective is only to prove that the cDNA from the KO sample is of good quality, amplifiable, and that therefore, the absence of Spata3 amplification is conclusive.

-We think that we already discussed the difference between in vivo and in vitro results in the discussion section (lines 220-226 in the modified version).

We extended the discussion concerning the percentage of abnormal sperm. First, a precision was added in the results section.

“In some extreme cases, sperm could also look “inflated”, showing an asymmetrical acrosome, completely untied from the nucleus, as exemplified in Figure 5a. A count of this particular phenotype revealed a greater abundance in KO (15%) compared to WT (3%) samples (Figure 5c). Some other sperm showed only one or the other defect, not taken into account in the quantitative evaluation. “

Then, in the discussion :

“It is difficult to estimate exactly what percentage of sperm is abnormal, as more than 10% show lipid droplets, and 15% appeared fully disorganized at the acrosome level, and some other showed milder defects. “

Reviewer 2 Report

The manuscript by Girault and colleagues is aimed to assess whether the Spata3 gene is involved in spermatogenesis and fertility in mice. To accomplish this, the authors use an in vivo model of KO mice, where they demonstrate the occurence of an abnormal acrosome ultrastructure, associated with altered in vitro fertilization assays. Overall, the manuscritp is well-written, and sufficiently detailed. The English is fluent. Methods fit with the authors' purpose. The only comment I would suggest is to add the analysis of data distribution. Indeed, the t-test has been performed, but no detail about the normal distribution of the data is provided. Another main limitation of this article concern the almost total lack of evidence of clinical translational aspect.

Author Response

We thank reviewer 2 for his/her pertinent comments.

- Indeed, after checking, some data were not fully normally distributed. We applied a Welch’s correction to the t-test without any consequence on the final result. This point was mentioned in the MM section.

“ When data were not normally distributed, a Welch’s correction was applied.“

-Indeed, it would be interesting to analyze the SPATA3 gene in infertile men. We already sequenced 3 samples from infertile patients with partial globozoospermia and in collaboration with another lab, 18 additional patients were investigated, for which their globozoospermia was not explained by anomalies of the DPY19L2 gene (unpublished data). No mutation could be detected.

At the end of the discussion, a sentence was added to open a perspective of clinical investigations :

“On the human side, SPATA3 and other potential candidate genes from the QTL region should be screened in infertile/hypofertile patients with signs reminiscent of globozoospermia. Furthermore, genes identified in this QTL region will be particularly analyzed in our future whole exome sequencing studies of infertile patients. “

Round 2

Reviewer 1 Report

During first round of the review, authors properly addressed my major reservations, I have 2 remaining points:

1) Figure 1 - thank you for updating the figure, but I consider it as still not convincing. Newly added panels g, h have visible lower intensity of the background compared to b, e and authors should reasure readers that same image acquisition parameters were applied on experimental/control samples. The magnified panel g contains only spermatogonia/spermatocytes (no sperm/ascrosomes). From Figure it is obvious for informed reader that authors do not have properly perfomed and interpreted IF acrosome antibody labeling in their hands and since authors further provided the info about the antibody issue, I suggest to remove the figure from the manuscript (point 2).

2) I thank author for the provided review of the antibodies marker situation. The overall info should be incorporated into the manuscript to explain to readers missing standard mice KO model characterization procedures. I am leaving the decision if the manuscript is eligible for publication in IJMS without immuno-western blot KO verification up to the journal editors.         

Author Response

We thank the reviewer and we fully understand his/her reservations.

Unfortunately, and as we already explained in the first round of the revision, the antibody used in this experiment is no longer commercially available to perform more experiments and to provide new convincing images at a time for the reviewer and especially for the reader.

We have good reasons to think that the labeling observed with the antibody is very specific and that the control carried out with the secondary antibody alone is indeed negative.

We now provide the raw images (In the ZIP file with the modified manuscript) to the reviewer so that he can get a better idea and maybe a conviction. Obviously, all images were taken with the same acquisition parameters and in particular, the same exposure.

Image 1: labeling with the antibody. Section including many spermatozoa.

Image 2: labeling with the secondary antibody alone.

Image 3 : labeling with the antibody. Section including many spermatids.

Moreover, we agree with the reviewer on the fact that the section proposed in the figure does not show spermatozoa in the lumen. Therefore, we show another image (#3, also raw) which shows that the signal also exists on less mature germ cells than spermatozoa and in particular on spermatids. This signal is absent when only the secondary antibody is used.

We hope that these images will eventually convince the reviewer. We therefore propose a new version of Figure 1 in the manuscript, because we consider that this information is useful for the readers. The text was modified accordingly.

We however remain completely open to further modify this figure according to the reviewer's advice and even to delete it, as was suggested in the last comment.

We included in the text a sentence mentioning the absence of antibody availability (lines 92 – 95 in the modified version):

As the antibody previously used for immunofluorescence detection (Figure 1) is no longer commercialized and no other validated antibody is available for Western blotting, the absence of expression of the Spata3 gene could only be checked by RT-PCR.

Reviewer 2 Report

Accept in this form

Author Response

We thank the reviewer.